# Bacterial Symbionts in *Ceratitis capitata*

**DOI:** 10.3390/insects13050474

**Published:** 2022-05-19

**Authors:** Alessia Cappelli, Dezemona Petrelli, Giuliano Gasperi, Aurelio Giuseppe Maria Serrao, Irene Ricci, Claudia Damiani, Guido Favia

**Affiliations:** 1School of Biosciences & Veterinary Medicine, University of Camerino, CIRM Italian Malaria Network, Via Gentile III da Varano, 62032 Camerino, Italy; alessia.cappelli@unicam.it (A.C.); aureliogiuseppe.serrao@unicam.it (A.G.M.S.); irene.ricci@unicam.it (I.R.); guido.favia@unicam.it (G.F.); 2School of Biosciences & Veterinary Medicine, University of Camerino, Via Gentile III da Varano, 62032 Camerino, Italy; dezemona.petrelli@unicam.it; 3Department of Biology and Biotechnology, University of Pavia, 27100 Pavia, Italy; gasperi@unipv.it

**Keywords:** *Ceratitis capitata*, *Asaia*, Symbiotic Control

## Abstract

**Simple Summary:**

The Mediterranean fly (Medfly), *Ceratitis capitata*, is considered one of the world’s most destructive fruit pests, as it can attack commercially important fruit, thus causing considerable economic damages, estimated to be more than 2 billion dollars annually. The yield reductions are mainly due to the damage incurred when larvae feed directly on the pulp, inducing the premature fruit drop. Additionally, oviposition holes facilitate secondary fungal and bacterial infections, further reducing the yields. Integrated pest management (IPM) strategies for medfly control are highly dependent on the use of insecticides, which, however, pose environmental concerns. Alternative strategies include the Sterile Insect Technique (SIT), which aims to eliminate or suppress pest insects without using pesticides. Lately, the medfly microbiota has been explored to develop new control strategies for insect pests and insect vectors. Here, we report the characterization of the microbial communities associated with selected organs of three different populations of *C. capitata* to identify possible candidates for a Symbiotic Control approach. Our findings provide new knowledge about the microbiota associated with *C. capitata* and stress the characterization of microbial symbionts as possible tools for a Symbiotic Control approach to implementing the pest management programs.

**Abstract:**

*Ceratitis capitata* (Diptera: Tephritidae) is responsible for extensive damage in agriculture with important economic losses. Several strategies have been proposed to control this insect pest including insecticides and the Sterile Insect Technique. Traditional control methods should be implemented by innovative tools, among which those based on insect symbionts seem very promising. Our study aimed to investigate, through the 16S Miseq analysis, the microbial communities associated with selected organs in three different medfly populations to identify possible candidates to develop symbiont-based control approaches. Our results confirm that *Klebsiella* and *Providencia* are the dominant bacteria in guts, while a more diversified microbial community has been detected in reproductive organs. Concertedly, we revealed for the first time the presence of *Chroococcidiopsis* and *Propionibacterium* as stable components of the medfly’s microbiota. Additionally, in the reproductive organs, we detected *Asaia*, a bacterium already proposed as a tool in the Symbiotic Control of Vector-Borne Diseases. A strain of *Asaia*, genetically modified to produce a green fluorescent protein, was used to ascertain the ability of *Asaia* to colonize specific organs of *C. capitata*. Our study lays the foundation for the development of control methods for *C. capitata* based on the use of symbiont bacteria.

## 1. Introduction

*Ceratitis capitata* (Diptera: Tephritidae) is one of the main destructive pests of fruit production worldwide [1] because of its significant physical damage to fruits and vegetables and its economic impact on agriculture and forestry. Due to its ability to tolerate and adapt to a wide range of climates and its capability to attack a broad spectrum of plant species, *C. capitata* has a large distribution. This is the case in Mediterranean countries as well, considering that medfly damage (linked to the population density) is more consistent in intensive cultures such as in Latin America than in Mediterranean agriculture [2]. Several strategies have been proposed to control the medfly distribution. Among them, methods such as the insecticide bait spray and the Sterile Insect Technique (SIT) have been demonstrated to be effective [3,4]. Although the chemical method works efficiently in medfly control, it has some disadvantages related to its toxicity to humans and animals and makes the fruits or plants polluted by leaving residues on them. Despite the fact that SIT has been successful in several countries, releasing X-rays sterile medflies aimed to reduce the wild insect population and helping to reduce the broad spectrum of chemical treatments, new control methods that are easily applicable are urgently needed.

Indeed, the study of microbiota could open new perspectives in medfly control to integrate other control methods. In fact, in several entomological systems (both insect pests and insect vectors), the use of symbiotic microbes to control harmful insects is already common practice in some cases; in others, it may become so in the near future [5,6,7]. A striking case in this context is represented by mosquitos’ vectors of pathogens that are dangerous for humans and animals. In particular, some selected bacteria have been enrolled in mosquito control. One very effective method is the biocontrol of mosquito borne diseases through mosquitoes transinfected with the endosymbiotic bacteria *Wolbachia*. Others refer to the bacterium *Asaia*, which has been isolated from many different mosquito species. *Asaia* is a natural effector for mosquito immune priming and is often detected at a very high prevalence, and this is attributed to several aspects of its biology. It has also been proposed for paratransgenic control applications [8,9].

Nevertheless, up to now, studies on the microbiota composition of *C. capitata* are already limited [10,11,12,13,14] and mostly related to the SIT and Incompatible Insect Technique (IIT). It has been noted that these procedures affect the medfly gut bacterial community, and sterile males are less competent in attracting and mating with wild females and are affected by the genetic background of the medfly population [15,16,17]. Post-irradiation dietary supplementation with *Klebsiella oxytoca* improved the mating latency of infertile males, suggesting the potential use of this bacterium to improve the SIT success [15]. Additionally, selected bacteria can act as probiotics in the diet of larval and adult stages, improving *C. capitata* mass rearing and enhancing the SIT application [18]. Similar to *C. capitata*, the intestinal probiotic *K. oxytoca* restored the ecological fitness of *Bactrocera dorsalis*, another destructive, polyphagous and invasive insect pest of fruits and vegetables, following a decline post-irradiation [19]. In *B. dorsalis* bacteria belonging to Enterobacteriaceae, *Enterobacter cloacae*, *K. oxytoca*, *Morganella* sp., *Providencia rettgerii* and *C. freundii* were identified as predominant in the gut, suggesting their involvement in promoting host fitness under stressful conditions and host immune system response [20,21,22]. *K. michiganensis* was implicated in promoting insect resistance to long-term, low-temperature stress [23]. Moreover, the gut symbiont *Citrobacter* sp. seemed to be involved in increasing the host resistance against an organophosphate insecticide [20].

Here, we present the results of a study aimed to characterize the microbial communities in different anatomical districts (guts and reproductive organs) of three different populations of *C. capitata* to determine whether selected symbionts could be translated into potential tools for the Symbiotic Control (SC) of medfly, which could be integrated with more traditional approaches. Considering the presence of *Asaia* in all three medfly populations, we investigate the possibility of using *Asaia* in the SC of *C. capitata.*

## 2. Materials and Methods

### 2.1. Ceratitis capitata Rearing

The strains of *C. capitata* used in this work were: (i) the Guatemala strain, established in 1989 from wild pupae collected in Antigua (Guatemala); (ii) the La Réunion strain, established in 1994 from wild pupae collected near St. Denis (La Reunion, France); (iii) the ISPRA strain, established in 1968 at the European Community Joint Research Centre (Ispra, Italy) with wild flies from Sicily and Greece and maintained in Pavia since 1979. These strains originated from the Department of Biology & Biotechnology, University of Pavia, where they are maintained under standard rearing conditions [24], and, since 2018, they have been maintained in the insectary at the School of Biosciences and Veterinary Medicine, University of Camerino. The insects were kept in cages (25 × 25 × 25 cm) made of a steel frame covered with nettings and maintained at the standard lighting conditions of 12 h light and 12 h dark, 26 ± 1 °C temperature and 75 ± 5% humidity and in aseptic conditions during both the developmental and adult stages.

### 2.2. DNA Extraction

Before DNA extraction, the insect surface was sterilized in 70% ethanol and rinsed three times in sterile PBS. The samples were homogenized with sterile 0.5-mm wide glass beads (Bertin Instrument, Montigny Le-Bretonneux, France) for 30 s at 6800 rpm by an automatic tissue homogenizer (Precellys 24 Bertin Instrument, Montigny Le-Bretonneux, France). Genomic DNA was extracted using a Jet Flex Genomic DNA Purification kit (Invitrogen, Thermo Fisher Scientific, Waltham, MA, USA), according to the manufacturer’s instructions.

### 2.3. 16S rRNA Profiling

The 16S Miseq analysis was conducted on the male and female organs (gut and reproductive organs) of three different *C. capitata* populations. Single pools of 20 organs for each group were obtained from cohorts of 10-day-old insects dissecting in sterile conditions. DNA extraction was performed as described above. A negative control consisting of a blank sample was included for each batch of extraction to control for the contamination of bacteria possibly introduced during the DNA extraction. They were not further processed since no quantifiable extract was produced from each negative control.

16S rRNA profiling was conducted by LGC Genomics (Berlin, Germany). Libraries preparation was performed by covering the hypervariable region V3–V4 of 16S ribosomal RNA using 341F and 785R oligonucleotides [25]. The data were pre-processed using the Illumina bcl2fastq 2.17.1.14 software, and the reads were sorted by amplicon inline barcodes. Sequencing adapter remnants were clipped from all reads. 16S pre-processing and OTU picking from the amplicons were analyzed using Mothur 1.35.1 [26]. The sequence alignments were performed against the 16S Mothur-Silva SEED r119 reference alignment. OTU diversity was analyzed with QIIME 1.9.0 [27], and annotations of the putative species level of OTUs were obtained with NCBI BLAST+ 2.2.29 [28]. The raw data were submitted as BioProject accession number PRJNA682004 to the NCBI database.

### 2.4. Molecular Detection of Asaia

The presence of *Asaia* was investigated in 20 female and 20 male adults from the three *C. capitata* populations. *Asaia* detection was performed by PCR using specific oligonucleotides targeting the 270-bps fragment of the 16S rRNA gene described in Favia et al. [8]. A total of 50 ng of genomic DNA were used in PCR reaction containing 1X Buffer, 0.25 mM dNTPs, 0.9 U DreamTaq Polymerase (Thermo Scientific, Waltham, MA, USA) and 200 nM of Asafor (5′-GCGCGTAGGCGGTTTACAC-3′) and Asarev (5′-AGCGTCAGTAATGAGCCAGGTT-3′) oligonucleotides. The amplification protocol included: initial denaturation at 95 °C for 3 min, followed by 30 cycles consisting of denaturation at 95 °C for 30 s, annealing at 60 °C for 30 s and extension at 72 °C for 30 s and concluding with a final extension step of 10 min at 72 °C. The PCR products were electrophoresed on a 1% agarose gel to determine the presence and size of the amplified DNA. The amplicons were purified and sequenced by the mean of the Sanger method (Eurofins Genomics, Ebersberg, Germany). The 16S RNA sequences were analyzed by BLASTN (http://blast.ncbi.nlm.nih.gov/Blast.cgi, accessed on 1 May 2022).

### 2.5. Asaia Isolation from C. capitata Adults

Bacteria isolation was performed from *C. capitata* adults. Before the isolation, the insect surface was sterilized in 70% ethanol and rinsed three times in sterile PBS. *Asaia* isolation was performed as described in Favia et al. [8] using a liquid enrichment medium at pH 3.5 followed by plating in a carbonate-containing solid medium. The enrichment medium allowed for the elimination of those microorganisms not tolerating low pH, while the following plating on CaCO_3_-containing medium permitted the isolation of acid-producing bacteria capable of creating CaCO_3_ dissolution haloes around the colonies. To confirm the isolation, pink-pigmented, shiny and smooth colonies were analyzed though *Asaia*-specific PCR.

### 2.6. Asaia sp. Transformation

*Asaia* strains isolated from the three populations of *C. capitata* were transformed with plasmids pHM2 containing a GFP gene cassette to express a green phenotype, as described in Favia et al. [8]. *Asaia* cells were grown overnight on GLY agar (yeast extract 1%, glycerol 2.5%, agar 2%, pH 5) for 48 h at 30 °C. The culture was diluted 1:20 into 25 mL of GLY and incubated with aeration until the cells reached the early log phase (optical density at 600 nm, 0.5–0.8). After an incubation of 15 min on ice, the cells were harvested (2700× *g*, 10 min, 4 °C) and washed twice with 10 mL of cold 1 mM HEPES (pH 7.0). The pellet was resuspended in 5 mL of cold 10% (*v*/*v*) glycerol, centrifuged again and finally resuspended in 0.5 mL of cold 10% (*v*/*v*) glycerol. Then, 65 microliters of competent cells (≈9.2 × 10^9^ CFU/mL) were mixed with 0.2 μg of plasmid DNA, transferred to a cold 0.1 cm-diameter cuvette and pulsed at 1800 V in the Electroporator apparatus (Biorad, Hercules, CA, USA). The cells were immediately added with 1 mL of GLY medium, transferred to a tube and incubated at 30 °C for 3 h. Transformant cells were selected by plating on GLY agar medium added with 100 μg/mL kanamycin, 40 μg/mL 5-bromo-4-chloro-3-indolyl-β-D-galactopyranoside (XGal) 24 and 0.5 mM isopropyl-β-D-thiogalactopyranoside (IPTG) for testing the Lac+ phenotype. Successful transformants were confirmed by fluorescence microscopy analysis and specific *Asaia*-PCR. A Gfp-tagged transformant from the ISPRA population was hence selected for mosquito recolonization experiments.

### 2.7. Medfly Recolonization by Asaia Expressing Green Fluorescent Protein (Gfp)

The ISPRA population was fed with modified *Asaia* to evaluate the ability of the bacterium to colonize the insect host. The *Asaia*^Gfp^ strain from the ISPRA medfly was grown for 24 h at 30 °C in GLY medium enriched with 100 µg/mL kanamycin. The cells were harvested by centrifugation, washed three times in 0.9% NaCl, adjusted to a concentration of 10^8^ cells/mL and resuspended in sugar solution 25%. For monitoring the long-term colonization of *C. capitata*, the suspension was supplemented with 100 μg/mL of kanamycin to avoid the loss of plasmid from the bacterial cells. After 3, 5, 10 and 15 days of exposure to the feeding solutions containing *Asaia*^Gfp^, the insects were dissected to monitor the presence of green bacteria in the organs. At each timepoint, the guts and reproductive organs of 7 males and 7 females were dissected in sterile PBS, fixed with 4% paraformaldehyde for 10 min at 4 °C and mounted in glycerol–PBS for analysis with a Nikon C2+ Laser Scanning Confocal Microscope (Nikon, Minato, Tokyo, Japan).

### 2.8. Isolation of Bacteria from the ISPRA Population

Bacteria from the ISPRA population medfly were isolated, processing a single pool of 10 gut and 10 reproductive organs from males and females. Each pool was homogenized as previously described in 100 µL of sterile 1X PBS, and serial dilution in 1X PBS was prepared from 1:10 to 1:1000. One hundred microliters of each dilution were spread on the surface of Blood Agar plates (ThermoFisher Scientific, Waltham, MA, USA) and incubated at 35 °C for 48 h. The isolated colonies were picked and transferred on Luria Bertani (LB) agar plates (1% NaCl, 1% Tryptone, 0.5% Yeast extract, 2% Agar, pH7). To isolate gram-negative bacteria, clones were plated on MacConkey agar (ThermoFisher Scientific, Waltham, MA, USA) and incubated at 35 °C for 48 h. The identification of gram-negative oxidase-negative isolates was performed using the API20E system (bioMérieux, Inc, Marcy-l’Étoile, France).

The other isolates were molecularly characterized using universal oligonucleotides targeting the 16S RNA ribosomal bacterial (805 Rev 5′-TCGACATCGTTTACGGCGTG-3′ and 27 for 5′-AGAGTTTGATCCTGGCTCAG-3′). A total of 50 ng of genomic DNA were used in PCR reaction containing 1× Buffer, 0.25 mM dNTPs, 0.9 U DreamTaq Polymerase (Thermo Scientific, Waltham, MA, USA), 200 nM of 27F and 805R oligonucleotides. The amplification protocol included: initial denaturation at 95 °C for 3 min, followed by 30 cycles consisting of denaturation at 95 °C for 30 s, annealing at 56 °C for 30 s and extension at 72 °C for 30 s and concluding with a final extension step of 10 min at 72 °C. The PCR products were electrophoresed on a 1% agarose gel to determine the presence and size of the amplified DNA. The amplicons were purified and sequenced by the mean of the Sanger method (Eurofins Genomics, Germany). The 16S RNA sequences were analyzed by BLASTN (http://blast.ncbi.nlm.nih.gov/Blast.cgi, accessed on 1 May 2022).

### 2.9. Antagonistic Assay

Antagonistic activity of the *Klebsiella oxytoca* strain was evaluated by an Agar diffusion assay against the *Asaia* strain isolated from the ISPRA population medfly as described in Ciolfi and Marri [29], with some modifications. Briefly, overnight cultures of *K. oxitoca* and *Asaia* in a GLY medium were used for the test. A lawn was prepared by mixing 5 mL of soft-agar (0.7% agar) with 59 μL of *Asaia* culture (OD600, 0.1) and poured over the GLY agar medium in plates. Three microliters of *K. oxytoca* (OD600, 0.3) were then spotted on the lawn. The plates were incubated for 4 days at 26 °C and examined daily for zones of inhibition.

## 3. Results

### 3.1. 16S rRNA Profiling

The microbiome sequencing of male and female organs of three different populations of *C. capitata* generated a total of 4.2 M reads, varying among samples (minimum = 64.760, maximum = 701.290), with an average of 352.944 reads. Analysis of the rarefaction curves indicated an adequate sampling quality, suggesting a coherent number of sequence reads per sample. The Principal Coordinates Analysis (PCoA) plots show the high similarity of microbial composition among all of the guts analyzed, while a more specific microbial community in the ISPRA reproductive organs was represented. No substantial difference in microbial composition is observed between males and females in any strain (Appendix A). At the phylum level, the Proteobacteria results are the most prevalent in all groups, particularly in guts (males and females 99%). The phyla Actinobacteria and Firmicutes were detected in reproductive organs. Bacteroides phylum was revealed in the reproductive organs of La Réunion (males: 2.7%) and ISPRA strains (male: 6% and females: 2.3%). Additionally, Cyanobacteria phylum was present in ISPRA population reproductive organs (males: 15% and females: 7.8%) (Figure 1 and Appendix A).

At the genus level, among the Proteobacteria phylum, *Klebsiella*, belonging to the class of Gammaproteobacteria, was the most abundant bacteria, with a range around 85–98% in the guts and 33–80% in the male and female reproductive organs of all three populations, except in the La Rèunion females, where *Providencia* is highly represented (95.7%), while *Klebsiella* is at 1.8%. *Providencia* was detected with a different percentage range (6–95%) in all samples—although in the male guts of the La Réunion strain and the female guts of the ISPRA strain, the percentage was lower than 1% (0.3% and 0.5%, respectively) (Figure 2 and Appendix A).

Other interesting bacteria belonging to the phylum Proteobacteria (class: Alphaproteobacteria) such as *Asaia* and *Gluconobacter* were also detected, which were present in all samples, albeit with variable percentages (see Appendix A Appendix A). Considering the 1% cut-off for sample analysis, *Asaia* was present in the male reproductive organs (1.1%) and female guts (1.4%) in the Guatemala strain, the male reproductive organs (8%) of the La Réunion strain and the male guts (1%) and male (7.4%) and female (8%) reproductive organs of the ISPRA strain.

*Gluconobacter* was mostly detected in the males (guts: 1% and reproductive organs: 3.2%) of the La Réunion population and in the male (3.6%) and female (3.2%) reproductive organs of the ISPRA strain.

Just in the ISPRA population, the bacterium *Chroococcidiopsis*, belonging to the phylum Cyanobacteria, reached a much higher density, being detected in male (14.9%) and female (7.6%) reproductive organs.

In all three populations, *Propionibacterium* was detected only in male and female reproductive organs, with a range of 1–20%.

As already reported in several studies, no *Wolbachia* was detected in any samples [30].

### 3.2. Detection, Isolation and Transformation of Asaia from C. capitata Populations

Considering the presence of *Asaia* shown in the NGS analysis, we investigated its presence in the new generation of medfly in depth. We tested for *Asaia* presence in 40 adults (20 males and 20 females) of each population by specific PCR. *Asaia* was detected in 100% of samples (a total of 120 individuals). Moreover, *Asaia* was isolated from *C. capitata* adults using a selected medium followed by plating in a carbonate-rich medium. The resulting single, pink colonies capable of dissolving carbonate in the medium and generating dissolution haloes were confirmed to be *Asaia* by 16S rRNA gene sequencing, with 99.8% nucleotide identity with *A. bogorensis* (Figure 3A–C). Additionally, *Asaia* was genetically modified to express a GFP protein to confirm its versability (Figure 3D–F).

### 3.3. Medfly Recolonization Using Asaia Strain Expressing Green Fluorescent Protein (Gfp)

To investigate the possibility of using *Asaia* as a Symbiotic Control tool for medfly, we tested its ability to colonize the insect’s organs in the ISPRA population, where *Asaia* was most abundant in the reproductive organs with respect to the other populations. *Asaia*-GFP was observed in the crop, an initial portion of the alimentary tract, of all samples (males: 7/7 and females: 7/7) only at the first time point (3 days after *Asaia* administration) (Figure 4). At the following time points, no individual tested positive for fluorescence. Likely, the presence of strong dominant bacteria, such as *Klebsiella* and *Providencia*, and the unfavorable sugary diet may have prevented *Asaia* from colonizing medfly organs in the long term.

### 3.4. Bacteria Isolation from the ISPRA Population and Activity Test

Considering the competition phenomena of symbionts to colonize specific niches in insects [31,32], we proceeded with the isolation of other bacteria from male and female organs (guts and reproductive organs) to select microorganisms showing potential features to interfere with the presence of *Asaia*. From each pool of guts plated on MacConkey Agar, *Klebsiella oxytoca* (Enterobacteriaceae family) and *Serratia marcescens* (Yersiniaceae) were isolated. The same species were found in the reproductive organs of both males and females while, *Aeromonas* spp. (Aeromonadaceae) was detected only in female reproductive organs. Moreover, the sequencing of a longer fragment of the16S rRNA gene (780 bps) of other isolates allowed for the identification of the bacterium *Enterococcus* sp. (Enterococcaceae), with an identity of 99% (MK764705.1) in the same organs.

Since *K. oxytoca* has been described to have a killer phenotype [29], we hypothesized that the strong presence of this bacterium in the gut could interfere with *Asaia* stabilization in this anatomical district. We tested the killer activity of *K. oxytoca* against *Asaia*, plating the tester bacterium in soft agar and adding a drop of *K. oxytoca* in the center. After 4 days at 26 °C, we observed a homogeneous layer of *Asaia* grown around the spot of *Klebsiella*, suggesting that *Asaia* was not sensible to the *K. oxytoca* activity (Appendix A).

## 4. Discussion

The microbiota composition of three different populations of *C. capitata*, which were reared under the same conditions, showed a homogeneous microbial community in the guts, which is dominated by *Klebsiella* and *Providencia*, as reported in previous studies [29,33,34,35], while highly different microbial communities were detected in the reproductive organs. These data support the need to investigate bacterial communities associated with different anatomical districts to identify potentially useful microorganisms in the development of medfly Symbiotic Control approaches. In this context, the first detection of *Chroococcidiopsis* and *Propionibacterium* as stable components of the medfly’s microbiota may open interesting perspectives. *Propionibacterium* has been already described as part of the microbiota of other insects, such as mosquitos [36], likely accomplishing a possible nutritional role, while *Chroococcidiopsis* has been never detected in insects, and its role in the reproductive organs is still unknown.

Among other interesting bacteria detected in our studies, we focused on *Klebsiella oxytoca* and *Enterococcus* sp, which were isolated from guts and reproductive organs. *Enterococcus* was previously described as part of the microbiota of *Bactrocera dorsalis*, where it seemed involved, together with *K. oxytoca*, in the reproduction and survival of the insect pest [37]. *Klebsiella* sp. has been previously described as a major component of wild medfly’s intestine [38], likely involved in nitrogen fixation [38,39] and associated with male mating success [40]. Additionally, *K. oxytoca* strains isolated from the alimentary tract of wild medflies showed an antagonistic activity in vitro against other bacteria such as *Escherichia coli*, *Enterobacter cloacae* and *Salmonella typhimurium* [29].

This is particularly interesting in the light of the fact that we detected *Asaia* in all of the samples but located only in the reproductive organs. In fact, *Asaia*, an insect symbiont proposed as a tool for the control of mosquito-borne diseases by direct paratransgenic applications and indirectly through the upregulation of the host immune response [9], has been described as strictly associated with several insects, such as mosquitoes, in association with salivary glands, guts and reproductive organs [8]. In some mosquito strains, *Asaia* competes with *Wolbachia* in the reproductive organs [41,42].

Thus, a possible competition between *Klebsiella* and *Asaia* could have been hypothesized. However, the strain of *Asaia* that we have isolated from *C. capitata* did not show susceptibility to *K. oxytoca* in the antagonistic assay. Although *Asaia* is resistant to antagonistic activity, we can hypothesize that interspecies competition among other bacterial populations for nutritional sources could explain the inability of *Asaia* to colonize the gut of medfly, but further investigations are needed to better understand these competitive phenomena.

## 5. Conclusions

Despite the growing interest in the role of the microbiota in the biology of many insects and in the possible use of microbiota components for the development of innovative control methods, only a few studies have been published to date on the microbiota of *Ceratitis capitata*. Our study, although still intended as a preliminary, has identified some bacteria that could potentially be used for the development of control methods based on symbionts and which, at the same time, offer the possibility of better understanding some aspects of microbial competition between the symbionts of insects.

## Figures and Tables

**Figure 1 insects-13-00474-f001:**
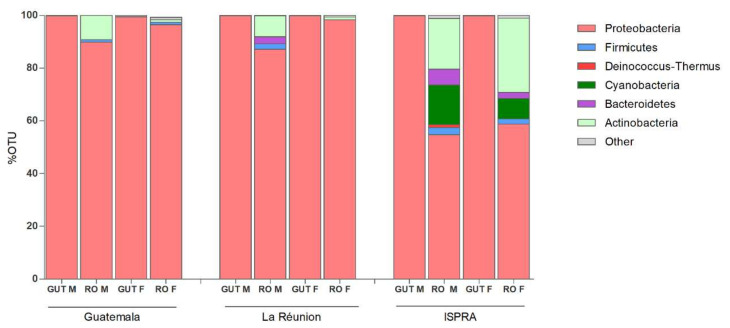
Phylum level composition (% of OTUs) in different organs of three different populations of *C. capitata.* Only OTUs representing >1% of the total reads are represented. RO: reproductive organs; F: females; M: males.

**Figure 2 insects-13-00474-f002:**
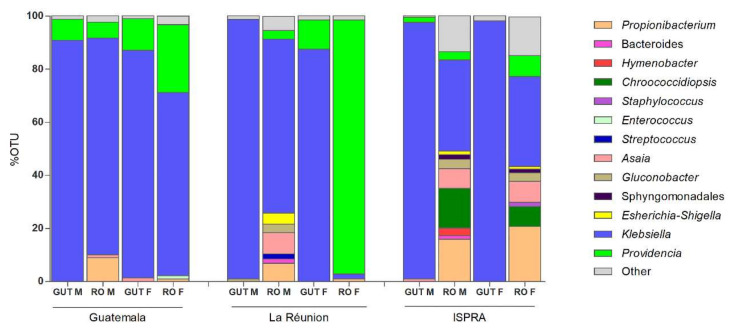
Genus level composition (% of OTUs) in different organs of *C. capitata.* Only OTUs representing >1% of the total reads are represented. RO: reproductive organs; M: male; F: female.

**Figure 3 insects-13-00474-f003:**
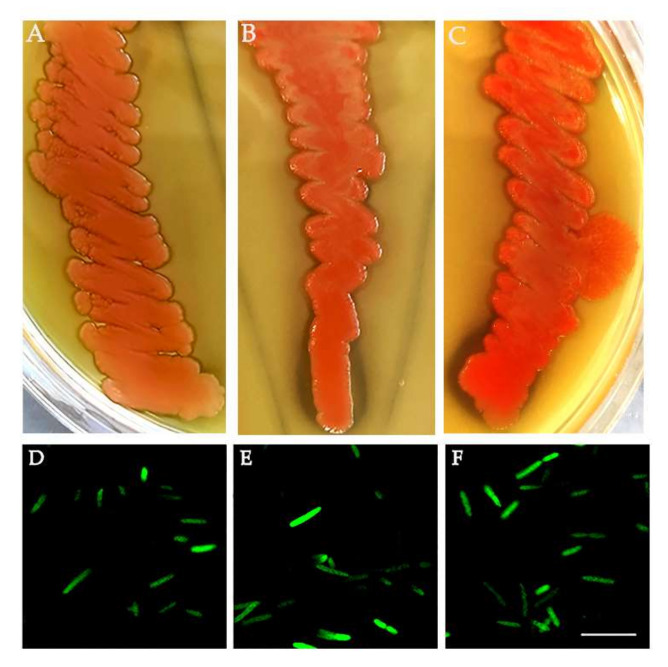
*Asaia* strains from *C. capitata*. *Asaia* isolates from the (**A**) Guatemala strain, (**B**) La réunion strain and (**C**) ISPRA strain. *Asaia* strains genetically modified to express a green fluorescent protein (GFP): (**D**) Guatemala strain, (**E**) La Réunion strain and (**F**) ISPRA strain. Bar = 10 µm.

**Figure 4 insects-13-00474-f004:**
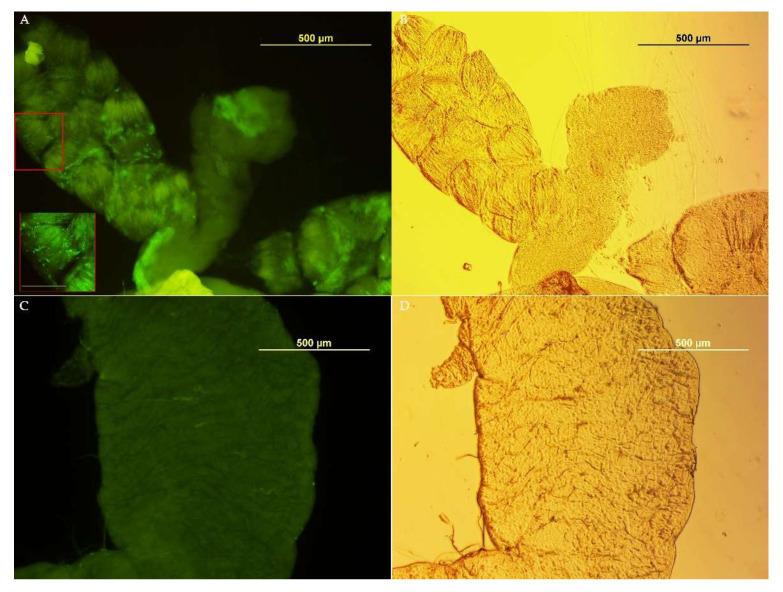
*Asaia*-GFP detection in *C. capitata* organs. (**A**) Gut colonized with *Asaia*-GFP; in the red box, a magnification of a gut portion with green bacteria (bar 50µm). (**B**) Phase-contrast image, (**C**) Gut from *C. capitata* fed on a standard diet (without *Asaia*). (**D**) Phase-contrast image.

## Data Availability

All the reads related to the 16S Miseq analysis (Bioproject PRJNA682004) have been deposited in The EMBL Nucleotide Sequence Database (NCBI).

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
