# Peer review of "Bacterial Symbionts in Ceratitis capitata"

_insects, 2022, doi:10.3390/insects13050474_

Round 1

Reviewer 1 Report

This paper shows  good progress in developing of possible tools for symbiotic control of the Mediterranean fruit fly approach implementing the pest management programs. 

Still, authors should give more data in introduction paragraph and list the similar research results on another Tephritidae fruit fly species (e.g. Bactrocera spp. microbial symbionts were extensively explored) and benefits of such research.  Therefore, the references list can be extended. 

Author Response

This paper shows  good progress in developing of possible tools for symbiotic control of the Mediterranean fruit fly approach implementing the pest management programs. 

We thank the Reviewer for the constructive comments

Still, authors should give more data in introduction paragraph and list the similar research results on another Tephritidae fruit fly species (e.g. Bactrocera spp. microbial symbionts were extensively explored) and benefits of such research.  Therefore, the references list can be extended. 

Consistent with the referee's suggestions, we integrated in the introduction a paragraph concerning Bactrocera dorsalis microbial symbiont composition and its possible role in host fitness (lines 80-89).

Reviewer 2 Report

This is a well-designed study with interesting results that deserve to published. In general, the manuscript is well written; however, I have some remarks that (I believe) could improve this paper:

Lines 27, 45 - please add "(Diptera: Tephritidae) after Ceratitis capitata

L98, 106 etc. - instead of "16S rRNA gene sequencing" it would be better to use "16S rRNA profiling" or at least "V3-V4 16S rRNA sequencing" to distinguish this approach from amplification and Sanger sequencing (?) of the longer (780 bp) 16S rRNA gene fragment. (The same concerns Results section).

L104-105 - actually, you should include them in PCR and NGS sequencing. There may have been DNA in small amounts that would have yielded some results in amplification and sequencing; the taxa detected could be properly accounted for in bioinformatic analyses.

L107 - did you sequence both V3 and V4 regions? (There is no a variable region between them).

L120 - instead of "targeting the 16S gene" it would be better to use "targeting xxx bp fragment of the 16S rRNA gene"

L124 - add space in "for3" (and between numbers and ng, ml etc.)

L192-199 - seems that it literally repeats information from 2.4 (L124-129)

L230 - genus should be in lowercase.

L319 and below - please remove "family" as the word ending "ceae" shows that this is family name.

L325 contains method ("were isolated and identified by APIWeb tool (bioMerieux)")

L328 - instead of "a portion of 16S RNA" should be "a longer fragment of the 16S rRNA gene".

L329 - homology is like pregnancy - it is or not (= to have a common ancestor). You should use "identities" or "99% of identical nucleotides"

There is no information how the longer amplicons were sequenced.

Author Response

This is a well-designed study with interesting results that deserve to published. In general, the manuscript is well written;

We thank the Reviewer for the constructive comments

however, I have some remarks that (I believe) could improve this paper:

Lines 27, 45 - please add "(Diptera: Tephritidae) after Ceratitis capitata

We have modified the text accordingly

L98, 106 etc. - instead of "16S rRNA gene sequencing" it would be better to use "16S rRNA profiling" or at least "V3-V4 16S rRNA sequencing" to distinguish this approach from amplification and Sanger sequencing (?) of the longer (780 bp) 16S rRNA gene fragment. (The same concerns Results section).

We have modified the text accordingly

L104-105 - actually, you should include them in PCR and NGS sequencing. There may have been DNA in small amounts that would have yielded some results in amplification and sequencing; the taxa detected could be properly accounted for in bioinformatic analyses.

We understand the referee's concern, but we wish to pinpoint that in our experiment we have included a negative control to evaluate contaminations during the DNA extraction process. No quantifiable extract was detected from the negative control; thus, no further processing was performed (in accordance with Alfano et al., 2019 (10.3389/fmicb.2019.02832)). 

The NGS analysis was performed by the LGC Genomics company which provided the raw data filtered from contaminations of PCR negative control specifically adopted by the company. 

L107 - did you sequence both V3 and V4 regions? (There is no a variable region between them).

We referred to paradigmatic works, consequently we sequenced the hypervariable V3 and V4 regions (Klindworth et al. 2013, 10.1093/nar/gks808). Primers we used, 341f/785r, produce the highest number of bacterial OTUs, phylogenetic richness, Shannon diversity, low non-specificity and most reproducible results (Thijs et al. 2017, 10.3389/fmicb.2017.00494).

L120 - instead of "targeting the 16S gene" it would be better to use "targeting xxx bp fragment of the 16S rRNA gene"

We have modified the text accordingly

L124 - add space in "for3" (and between numbers and ng, ml etc.)

We have modified the text accordingly

L192-199 - seems that it literally repeats information from 2.4 (L124-129)

We described the two PCR protocols since some of the parameters are quite different (i.e. annealing temperature)

L230 - genus should be in lowercase.

We have modified the text accordingly

L319 and below - please remove "family" as the word ending "ceae" shows that this is family name.

We have modified the text accordingly

L325 contains method ("were isolated and identified by APIWeb tool (bioMerieux)")

We have modified the text accordingly

L328 - instead of "a portion of 16S RNA" should be "a longer fragment of the 16S rRNA gene".

We have modified the text accordingly

L329 - homology is like pregnancy - it is or not (= to have a common ancestor). You should use "identities" or "99% of identical nucleotides"

We have modified the text accordingly

There is no information how the longer amplicons were sequenced.

Consistent with the referee's suggestion, we have provided further info about sequencing methods (lines 166 and 236).

Reviewer 3 Report

The authors use different approaches to describe the bacterial communities in several organ groups of the Mediterranean fly. The organ groups and how organs are pooled is hard to follow, but overall by checking the 16S there is a clear difference between reproductive organs and gut. Populations did not differ much, excepting the ISPRA group. Info at Fig 1 represent a very coarse level of resolution and are not really interesting. Lines 241-255 are really hard to follow. Overall, the writing style is not very polished, e.g. abuse of linkers and repetitions, and that clearly affects readability. See repetition of Although to start 2 consecutive sentences at 1st paragraph, same with In fact at the second. The Introduction is the most weak part in the sense that that not introduce the reader to anything that will find later. I am still not sure how the description of such communities will help to control the pest in the field, that link is assumed but not explained at all. Same with the whole potential of the study. The introduction also doesn't help the reader to understand all the different tests with Asaia, for instance. the biological question is sort of unclear, and therefore the efforts done later does not seem to have a clear goal. The section 3.3 contains tons of stuff that are not results. 

Some more comments in detail:

L91: ethanol and PBS will not degrade any DNA from surface, therefore the cleaning success is doubtful.

L100 on: the pooling/class/group concept could not be less clear. i had to check figures to find out...

L258: where is this potential coming from? If lit, this can not appear in results...

L311 the lack of long-term colonization by ingested cells is well known, due to the high renovation rate of gut epithelium...that is why most diet symbionts are transient. 

Author Response

The authors use different approaches to describe the bacterial communities in several organ groups of the Mediterranean fly. The organ groups and how organs are pooled is hard to follow, but overall by checking the 16S there is a clear difference between reproductive organs and gut. Populations did not differ much, excepting the ISPRA group.

We thank the Reviewer for the comments

Info at Fig 1 represent a very coarse level of resolution and are not really interesting. Lines 241-255 are really hard to follow.

The scheme we have adopted is a "classic" scheme in the representation of the composition of the microbiota of a host and follows the methodological scheme we adopted in our work. Fig 1 represents the variability in the composition of phyla while fig. 2 the variability of genera. The information in Fig 1 may seem to represent a very coarse level of resolution, but in our opinion, they are interesting as they show that at the phyla level there is no great variability while at the genera level the variability is substantial. We consider this an important element in the characterization of the microbiota in different medfly populations.

Moreover, we have rephrased the paragraph in order to make it clearer and more usable to the readers (lines 282-300).

Overall, the writing style is not very polished, e.g. abuse of linkers and repetitions, and that clearly affects readability. See repetition of Although to start 2 consecutive sentences at 1st paragraph, same with In fact at the second.

We have modified the text accordingly

The Introduction is the most weak part in the sense that that not introduce the reader to anything that will find later. I am still not sure how the description of such communities will help to control the pest in the field, that link is assumed but not explained at all. Same with the whole potential of the study. The introduction also doesn't help the reader to understand all the different tests with Asaia, for instance. the biological question is sort of unclear, and therefore the efforts done later does not seem to have a clear goal.

We substantially modified the introduction, accordingly with referee’s suggestion (lines 62-72 and 103-104).

The section 3.3 contains tons of stuff that are not results. 

We agreed with the reviewer and consequently we have much shortened this part (line 349).

Some more comments in detail:

L91: ethanol and PBS will not degrade any DNA from surface, therefore the cleaning success is doubtful.

We followed the previous protocol aimed to eliminate possible external surface contamination, not to degrade any DNA contamination (Mancini et al., 2018 doi: 10.1186/s12866-018-1266-9.; Alfano et al., 2019 10.3389/fmicb.2019.02832)

L100 on: the pooling/class/group concept could not be less clear. i had to check figures to find out...

We have modified the text accordingly (line 131)

L258: where is this potential coming from? If lit, this can not appear in results...

We have rephrased the sentence

L311 the lack of long-term colonization by ingested cells is well known, due to the high renovation rate of gut epithelium...that is why most diet symbionts are transient. 

With our long-lasting experience with Asaia studies, we recolonised several insects with fluorescent Asaia and we constantly observed a persistent long-term recolonization in all insects tested (Favia et al., 2007; Damiani et al., 2008; Crotti et al., 2009; Mancini et al, 2016). Likely, as suggested by reviewer 3, in C. capitata, Asaia seemed to be a transient bacterium. However, since we are unable to detect it into the intestine at 3-day post-infection, this supports the hypothesis that Asaia cannot persist in the midgut of medfly. In other insects, such as the Ae. albopictus mosquito, we observed an accumulation of Asaia in the hindgut which we do not observe in Ceratitis.

Round 2

Reviewer 3 Report

The authors made an effort to incorporate my previous comments. Being the result not spectacular in terms of quality, I still appreciate the novelty and interest to readers